# Different Discussion Partners and Their Effect on Depression among Older Adults

## Keunbok Lee

UCLA Department of Public Health, University of California, Los Angeles, CA 90095, USA; keunboklee@ucla.edu

**Abstract:** Although the multidimensionality of core discussion networks has been well established and widely studied, studies of the effects of social support on depression rarely consider the multifaceted aspects of dyadic discussion partner ties. This article proposes defining dyadic social relationships as a construct comprising several tie-level attributes and differentiating multiple forms of support relationships by assessing the configuration pattern of multiple attributes. The current study examines various forms of older adults' discussion partners and identifies which form of discussion partner relationship is effective at buffering the negative effects of adverse life events on depression symptoms. Results from the University of California Social Network Survey show that older adults' discussion partners can be classified into five distinct types of dyadic ties: spouse/romantic partners, close neighbors, remote type, social companions, and acquaintances. The discussion network with more close neighbor confidants is more effective at buffering the negative effects of adverse life events. These results offer an alternative way of investigating the differential significance of various social support relationships in mental well-being.

**Keywords:** multiple forms of discussion partners; depression; buffering effects of social support network; multilevel latent class analysis

## 1. Introduction

Social support networks provide health and survival benefits to old adults by maintaining social integration and strengthening coping under the adverse life events. Numerous studies have demonstrated that older adults with more supportive and integrated networks have better mental health and a lower risk of morbidity and mortality (Berkman and Syme 1979; Cohen 1985, 2004; House et al. 1988; Kaplan et al. 1977; Uchino et al. 2018). Despite the use of diverse definitions and measurements in previous research, there is a general agreement that social support is a multidimensional concept consisting of structural (e.g., size or interaction frequency), functional (e.g., types of support), and relationship (e.g., source of supports) dimensions (Cohen 1985; House et al. 1988; Thoits 1982, 1985; Wellman and Wortley 1990). Empirical studies have carefully interpreted the different mechanisms between social support networks and health according to the different dimensions of social support networks. For example, the functions of social support (e.g., emotional aid and instrumental aid) are often linked to a stress-buffering mechanism in which the negative effects of adverse life events on mental health are mitigated by social support. The effects of the support networks' structural aspects (e.g., size and contact frequency) on health, on the other hand, have been interpreted as the positive consequence of normative health-related regulations and social integration (Berkman et al. 2000; Kawachi and Berkman 2001).

Yet, despite its importance, the multidimensionality of social support is often conceptualized and operationalized only at the network level, without considering the multifaceted characteristics of dyadic support ties. A dyadic social relationship has several attributes such as interaction frequency, intimacy, and types of support. Given the fact that dyadic ties have multiple attributes, it is not surprising that each social tie takes its own relationship form (Adams et al. 1998; Allan 1998; Pahl 2000; Spencer and Pahl 2006). For

example, a close friend who gives emotional and instrumental help is quite different from a more distant friend who only gives instrumental help, even though both people are friends or provide instrumental help. Defining the complexity of dyadic relationships and the systemic assessment of the multiple forms of dyadic social relationships is necessary for understanding the potential importance of diverse, supportive social relationships in individual lives.

In this article, I extend previous research by examining the role of multiple forms of core social ties in buffering the effects of stress on depression symptoms. Specifically, this study examines core discussion partner networks, a relationship term widely used in social support network studies (Bookwala 2017; Brown and Harris 1978; Burt 1987; Cornwell 2012; Murphy 1982; Shiovitz-Ezra and Litwin 2012). Although core discussion partners are believed to be most supportive social relationships (Burt 1984; Marsden 1987; McPherson et al. 2006), recent studies have shown that discussion partner relationships vary greatly by the subject of discussion, individual circumstances, and relationship contexts (Bearman and Parigi 2004; Small 2013, 2017). Different core discussion partners provide different types of support (Brashears 2014). This study aims to classify multiple forms of discussion partner ties empirically and to examine what types of core discussion partner are most important in the stress-buffering role of social networks.

This study used rich personal network survey data collected from the San Francisco-Oakland area in 2015. The analysis is conducted in three steps to (1) classify the various forms of discussion partner ties based on multiple tie-level characteristics; (2) cluster the discussion partner networks at network level based on the multiple types of ties; then (3) examine interaction effects between adverse life events and discussion network typology on depression. The results of this study demonstrate five different types of discussion partner ties. They also show that close neighbor discussion partners seem to be crucial for buffering the negative effects of adverse life events on depression.

### 1.1. Multidimensionality of Social Support Networks

Although diverse definitions have been suggested, a social support network is generally understood to be a set of members of a personal network who provide emotional aid, instrumental aid, and informational aid (House et al. 1988; Kaplan et al. 1977; Thoits 1982). Numerous studies have shown the significant effects of social support upon psychological well-being, physical morbidity, and mortality (Berkman and Syme 1979; Cohen 1985; Seeman 2000). Belonging to stable social networks directly promotes positive psychological states, such as feelings of self-worth and social integration. Normative regulation of health-related behaviors exercised by social network members can reduce the likelihood of poor health behaviors (Berkman and Kawachi 2000; House et al. 1988; Kawachi and Berkman 2001; Umberson 1987). Network members' support buffers the negative impacts of stressful adverse conditions on mental health status. Discussing problems with network members helps redefine the importance of the issues and prevents maladaptive reactions to stressful events. In addition, the mobilized support may directly help individuals resolve the difficulties (Cobb 1976; Cohen 1985; Lin et al. 1985).

Though the operationalization of social support in empirical studies varies by survey instruments and research interests, a common way of measuring social support is to sum up the characteristics of social relationships. For example, the size of the network is a sum of the existing social ties. Emotional support is often measured by counting the total number of social network members who provide this support. Although aggregate measures are informative, additive measures rely on the crude assumption that the observed supports may be equally important. For example, when a study finds that having more emotional support significantly reduces depression, the association between emotional support and depression assumes that adding each emotional support reduces depression to the same degree as that which was achieved from adding the first emotional support. This symmetric linear association is possible only when every emotional support has an equally important effect against depression. However, as Thoits notes (Thoits 1982, p. 147), "not all sources or

types of social support are equally effective in reducing depression." The types of support people get from alters vary by the relationship (Brashears 2014; Wellman and Wortley 1990), and the effects of social support on psychological status depend on the characteristics of the relationship (Dean et al. 1990). For example, support from kin or adult children may exert weak effects on mental health because obligation strains independence and autonomy (Dean et al. 1990; Silverstein et al. 1996). Further, the social networks' benefits may be attributed mainly to supports from alters who have specific knowledge related to the ego's stressful situation (Perry and Pescosolido 2015). There is substantial evidence that the multidimensional aspects of social support and relationships are associated with the effects of social support on health outcomes.

The multidimensionality of social support networks has received considerable attention in social network and health studies. The literature has conceptualized three different dimensions of social support: the structures (e.g., size and density of networks), functions (e.g., types of support and social influence), and sources of social support (e.g., role relationships) (House et al. 1988; Kaplan et al. 1977; Thoits 1982; Baruch-Feldman et al. 2002; Dean et al. 1990; House et al. 1988; Thoits 1982; Wellman and Wortley 1990). Empirical studies often developed support measurements and interpreted the benefits of social support by distinguishing the specific dimensions of social support. For example, structural aspects of support networks often link to the social influence explanation, whereas the types of support people receive from their networks are interpreted as buffering factors that reduce the adverse effects of stress (Kawachi and Berkman 2001).

Instead of separating social support dimensions, network typology studies have suggested an alternative approach that identifies different types of support networks through complex combinations of structures, functions, and source of social support (Agneessens et al. 2006; Fiori et al. 2006; Litwin 1998; Litwin and Stoeckel 2014; Shiovitz-Ezra and Litwin 2012; Youm et al. 2018). The network typology research demonstrates that there are a countable number of social network types (usually between four and six), which differ from each other in terms of the proportion of support types (e.g., emotional support and instrumental aids), average size, and interaction frequencies, and distribution of role relationships.

Despite their contributions, studies taking multidimensionality into account have limited their attention to the complexity of support networks. The theoretical claim about multidimensionality can apply not only to the multidimensional aspects of support networks but also to the complexity of dyadic support relationships. For example, one observed dyadic support relationship can be described by applying the structural, functional, and relationship attributes of social support, such as a frequently interacting friend who provides emotional and instrumental aids. Although the concepts and operational strategy for the network-level multidimensionality have been well established, more conceptual and operational measures are needed to capture the tie-level complexity.

This study suggests conceptualizing the dyadic social tie as a multifaceted composite. Specifically, I argue that interactions of multiple attributes, including functions, structures, and other relationship-level characteristics, constitute the form of a dyadic social relationship. According to the pattern of associations among various attributes, diverse social ties take distinctive forms of relationships. The social support network in this study is redefined as a set of social relationships that takes multiple forms of relationship according to the configuration of its functional, structural, and other attributes. Examining the effects of social support on health by assessing the multiple forms of dyadic social ties is particularly beneficial to the asymmetric problem of the aggregated social support measures. As I discussed above, not every support or alter is equally important. Instead, the effect of one additional support (or alter) would depend on the types, functions, and importance of the relationship. Having a close friend who provides various types of support may not have an equal effect on health as having a more distant friend who only offers informational aid. By assessing the multiple forms of social ties, this study examines what forms of social

relationships are essential for understanding the stress-buffering role of social support networks.

*1.2. Diversity of Discussion Partners*

In assessing the buffering effect of multiple forms of a dyadic social relationship on depression, this study examines core discussion networks, which have been widely used in social network and health research. The social network members that people rely on for discussing important matters are thought to be the closest social relationships (Burt 1984; Marsden 1987; McPherson et al. 2006). Accordingly, having a small number of core discussion partners has been treated as a proxy of social isolation, weak social support, and difficulty of support mobilization, which in turn results in negative individual-level outcomes such as lower happiness (Burt 1987), bad self-rated health (Cornwell and Waite 2009), alcohol abuse, and physical inactivity (Shiovitz-Ezra and Litwin 2012).

However, recent studies on core discussion networks have shown that the discussion partners are neither homogenously close nor provide similar supports. People choose discussion partners from various social relationships based on discussion topics (Bearman and Parigi 2004), availability, and individual contexts (Small 2013, 2017). Bearman and Parigi (2004) found that particular discussion subjects are likely to be matched with particular role relationships. For example, a spouse is likely to be approached for economic and house-related issues, whereas friends would be partners for discussing community or ideological issues. Indeed, people sometimes discuss their important matters with newly encountered people in their current institutional contexts. For example, Small (2017) demonstrated that a substantial portion of graduate students' old and close discussion partners was replaced by newly encountered people such as roommates or administrators in their new institutional environment. Regardless of the emotional attachment, people disclose their personal matters to newly formed relations or acquaintances when they perceive that they have relevant knowledge and are available at the time of need (Small 2009, 2013, 2017). The support provided by discussion partners also differs depending on the role relationship and characteristics of alters. Using nationally representative data, Brashears (2014) showed that discussion topics, role relationships, and expected support are all significantly associated. For example, a spouse with whom one discusses economic issues is more likely than friends to provide monetary support (Brashears 2014).

The diversity of discussion partners implies that the social networks' buffering function varies according to what kind of discussion partners constitute the core discussion networks. One discussion network may be better for reducing stress under adverse circumstances than other networks, if the former network was more accessible, more knowledgeable, or had more supportive alters than the latter. Strong social relationships may successfully lessen the adverse effects of negative life events. People are likely to meet their basic psychological needs such as approval, esteem, or affirmation through the interaction with primary network members (Antonucci and Akiyama 1987; Kaplan et al. 1977; Thoits 2011). For example, having a confiding relationship with a marital partner may be the most effective factor in mitigating the adverse psychological impact of stressful life events (Brown and Harris 1978; Dean et al. 1990).

By contrast, some studies emphasize the role of secondary or weak ties in protecting mental health. Secondary social relationships may contribute to buffering stressors by suggesting fresh perspectives and information. For example, Perry and Pescosolido (2015) demonstrate that it is not total network size but the number of discussion partners people "talk to about health problems when they come up" that significantly improves individuals' mental health and health-related service satisfaction. While people tend to form close and supportive relations with others who they perceive to be similar to them (McPherson et al. 2001), the heterogeneous social relations in terms of gender, race/ethnicity, and aging also can offer benefits to protecting mental health. For example, cross-sex relationships in older adults provide a buffer against loneliness (O'Connor 1993). Network typology studies also show that a social network with diverse role relationships is more beneficial for reducing

depression than networks dominated by kin ties (Fiori et al. 2006; Fiori and Jager 2012; Litwin 1998; Litwin and Stoeckel 2014; Shiovitz-Ezra and Litwin 2012).

In sum, relationships with core discussion partners take heterogeneous forms according to relationship dimensions, including intimacy, accessibility, types of support, and role relationship. Accordingly, the core discussion network includes various forms of social relationships. In this case, the association between mental health and core discussion networks is not merely attributable to the amount of help and resources provided by close social relationships. Instead, the discussion networks' buffering effect on depression varies according to what forms of discussion partner relations compose the discussion network. Based on this discussion, I hypothesize that at the tie-level, there will be diverse discussant ties who take different forms of relationship in terms of intimacy, accessibility, and support provision. At the network level, the core discussion networks will be differently configured according to the distribution of multiple forms of discussants. Finally, I expect that the buffering effects of the core discussion networks on depression will vary according to the network configuration.

## 2. Data and Methods

### 2.1. Study Sample

I used the first wave of the University of California Berkeley Social Network survey data (UCNets) to examine variations in older adults' discussion partner relationships and their effects on mental health. The first wave of UCNets' respondents was drawn from Bay Area residents aged 20–30 and 50–70 years old in 2015. Survey respondents were selected from a random sample of households in 30 San Francisco Bay Area census tracts who responded to the solicitation letter and met the age criterion. In total, 1159 respondents (674 50–70 year-olds and 485 20–30 year-olds) completed the survey. This study is based on 568 older adults (50–70 years old) who reported at least one discussion partner and who completed questions on depression.

The main advantage of the UCNets is that this survey collected data on social networks and their members by asking several name-generating and -interpreting questions, which allowed the researcher to collect more reliable information than using a few name-generating questions (McCallister and Fischer 1978). The survey first asked respondents to list names of spouses, romantic partners, and housemates. Then respondents provided additional names of their network members through seven name-generating questions, specifically from whom did respondents (a) seek advice to make an important decision, (b) confide in regarding personal matters, (c) get practical help, (d) provide help to, (e) expect help from in a health crisis, (f) socialize together with, and (g) feel were demanding or difficult. The respondents were allowed to fill in multiple names on each question without restricting duplication. Based on the list of alters, the survey asked several name-interpreting questions that described the details of the alters and the relationship with them, such as emotional and geographical closeness, role relationships, homophily in age, gender, race, religion, and contact frequency.

The discussion partner in this study is defined using two name-eliciting questions: "When you have to make important decisions—for example, about taking a job, family issues, or health problems—whose advice do you or would you seek out?" and "Sometimes personal matters come up that concern people, like issues about relationships, important things in their lives, or difficult experiences. Who do you confide in about these sorts of things?" Respondents provided multiple names in these questions, and some alters appeared in both questions. I treated the alters who were named in at least one of these two questions as a discussion partner. Table 1 shows that 2487 alters were named as discussion partners, and the average number of discussion partners is 5.26.

**Table 1.** Descriptive statistics.

| Ego-Level Characteristics (*N* = 568) | | | Alter-Level Characteristics (*N* = 2487) | | |
|---|---|---|---|---|---|
| **Variables** | | **%** | **Variables** | | **%** |
| Gender | Male | 45.65% | Intimacy | Especially close | 68.33% |
| | Female | 54.35% | | Socializing | 58.07% |
| Marital status | Married/partner | 75.10% | Social exchanges | Sick help | 41.31% |
| | Never/divorced | 24.90% | | Practical help | 18.21% |
| Race/ethnic groups | White | 57.35% | | Provide help | 45.51% |
| | Non-white | 42.65% | | Living together | 16.83% |
| Education level | Less than college | 52.16% | Proximity | Within 1-h drive | 58.51% |
| | College | 26.96% | | More than 1-h drive | 24.66% |
| | More than college | 20.88% | | Spouse/romantic partner | 16.55% |
| Employment status | Not employed | 65.31% | | Kin | 32.74% |
| | Fully employed | 34.69% | Role relationship | Friends | 30.42% |
| Income level | Less than 35 K | 17.09% | | Coworkers | 7.56% |
| | 35–75 K | 25.85% | | Neighbors/group members | 9.54% |
| | More than 75 K | 57.06% | | Acquaintance | 3.19% |
| Survey mode | Face-to-face | 77.04% | | Same age | 50.55% |
| | Web interview | 22.96% | Similarity | Same sex | 63.97% |
| Mean (min/max) | | | | Same race/ethnic | 74.88% |
| Age | | 59.31 (50/70) | Newly-met | | 3.78% |
| Number of discussion partners | | 5.26 (1/15) | | | |
| Number of close discussion partners | | 2.74 (0/9) | | | |
| Number of adverse life events | | 0.95 (0/4) | | | |
| Depression | | 4.92 (0/21) | | | |
| Health status | | 2.48 (1/5) | | | |

## 2.2. Variables

I used the six tie-level profiling variables to classify the subtypes of discussion partners inductively; emotional closeness, geographical proximity, and four types of social exchange (socializing, sick help, practical help, and provide help). In order to further describe characteristics of subtype discussion partner ties, I used additional tie-level variables that capture the role relationship, similarity in gender, age, and race/ethnicity, and whether each tie was a newly formed relation or not. At the individual level, this study used 11 variables, including respondents' socio-demographic characteristics. The descriptive statistics for tie- and respondent-level variables are shown in the left and right panels of Table 1, respectively.

### 2.2.1. Tie-Level Variable

I used three main relationship components to profile the different types of discussion partners: intimacy, geographical reachability, and social exchanging. Intimacy was measured by a question about who the respondents feel especially close to. Geographical proximity was used as a proxy of reachability (Live together, Live at one-hour driving distance, or Live at more than one-hour driving distance). For the exchanging dimension, five variables were used: whether an alter was named as a social companion partner (Socializing), a person who provided help for practical chores (Practical help), a person for whom the respondent provided help (Provide help), and a person who respondents can rely on when respondents were sick (Sick help).

Five additional alter-attribute variables were used to describe the subtype of the discussion partner further: role relationship (spouse/romantic partner, kin, friends, coworker, neighbor/social organization members, acquaintance), a person first met in the last year ("yes" or "no"), and three similarity variables (same age, same-sex, and same race/ethnicity). In the initial survey, the respondents described the role relationship of an alter with more than ten categories and were allowed to choose multiple roles. I simplified the role relationship category in this study into six roles. "Spouse/romantic partner" represents either a married partner or person in a romantic relationship. "Kin" contains parents, children

(including step-relationships), siblings, in-law relationships, and other relatives. "Friends" are alters who were described as friends without any additional roles. For example, when a respondent defined a certain alter as a coworker as well as a friend, they were treated as a coworker. The "Neighbors or social organization members" category includes comembers of a religious or social club as well as neighbors. "Acquaintances" contains other relationships such as professionals (e.g., physician and landlady) or alters described as acquaintances.

As shown in the right panel of Table 1, not all discussion partners in the survey were emotionally or geographically close. While 68.3% of discussion partners were emotionally intimate, 31.7% were not. And 24.7% lived further than one-hour driving distance. Fewer than half of discussion partners were named social support exchanging partners, and only slightly more than half (58.1%) of discussion partners were identified as social companions. The role relationships also show the diversity of discussion partners. While the kin and friend groups take a large portion of discussion partners (32.7% and 30.4%, respectively), about one-fifth of discussion partners are coworkers (7.56%), neighbors or social organizations (9.54%), or acquaintances (3.19%). These simple statistics in the survey suggest the heterogeneity of relationships with discussion partners.

### 2.2.2. Respondent-Level Variables

I used a set of respondent-level variables for testing the interaction effects of adverse life events and discussion partner networks on depression symptoms. In this stage of analysis, I regressed depression symptoms on the number of negative events and the types of discussion networks derived from the previous profiling analysis as controlling other respondent-level characteristics.

Depression: The K6 version of the Center for Epidemiological Studies Depression Scale (K6-CES-D) was used to measure depression symptoms (Cole et al. 2004; Kessler et al. 2002): "feel restless or fidgety," "feel hopeless," "feel nervous," "feel so depressed," "feel that everything was an effort," and "feel worthless." Respondents reported how often they experienced these depressive symptoms during the past 30 days with five-point scales from "All of the time (0)" to "None of the time (4)." The original scales of these variables were reverse-coded and summed into a total score of depression (Cronbach's alpha = 0.761). A larger depression score represents a higher level of depression.

Number of negative events: UCNets asked a set of questions that captured whether respondents experienced various life events in the year before the survey. I selected four questions about life events that would negatively impact on depression: "Have you had any major problems at work?" "Have you had trouble paying your bills?" "Did anyone you felt close to pass away?" and "Has there been any major break in a relationship between yourself and a relative or close friend?" The number of adverse events was measured by counting the adverse events to which respondents answered "yes."

Control variables: This study used eight demographic characteristics (age, gender, marital status, race/ethnicity, education level, employment status, household income, and general health status), and two network variables (number of discussion partners and number of close discussion partners) as control variables at the second stage of analysis. These variables were also used for profiling the respondent's level of confidant network composition. The descriptive statistics of respondent-level variables are presented in the left panel of Table 1.

### 2.3. Analysis Strategy

The main analytical subjects of this study were as follows: first, inductively generating diverse composite portraits of discussion partner relationships; second, clustering discussion partner networks based on the varying composition of different types of discussion partners at the respondent level; and third, examining the buffering effects of different discussion partner networks against the negative effects of adverse life events on depression. Overall analysis was conducted with three separated steps using multilevel latent

class modeling. In the first step, the discussion partner ties were clustered into a number of subtypes of discussion partner relationships based on the association pattern of six profiling relationship variables and five tie-level covariates. Then in the flow analysis, I inductively grouped the respondents based on the distribution of subtypes of discussion partners. At this stage, the groups of respondents presented different types of discussion networks. By contrast, the cluster of discussion ties at the former step of analysis captured the different forms of discussion partners. After fixing the number of subtypes of discussion ties and discussion networks, in the last analysis step, I examined what types of discussion partner networks would be better than other types for mitigating the negative effects of unpleasant life events on a depression symptom.

I used the multilevel latent class model (MLLCA) for clustering discussion partners and discussion partner networks. The MLLCA is an advanced latent class analysis that classifies the study units into a certain number of subclusters based on observed variables' association patterns (Goodman 1974; McCutcheon 1987). MLLCA is well-suited for this study for three main reasons. First, MLLCA made it possible to address the biases that may result from the dependency of alters in estimating clusters with discussion partners. The ego-centric network survey data have a typical multilevel data structure in which alters, or ego−alter ties (lower units) are nested in the ego or ego-networks (upper units) (van Duijn et al. 1999; Snijders et al. 1995). The dependency of alters listed by the same respondent (in this study, discussion partners belonging to the same individual) leads to biased estimations in clustering lower-level units. The MLLCA accounts for the dependence among alters by introducing a random coefficient into the model that takes different values for each ego (Vermunt 2003).

The second advantage of MLLCA is that it allows the estimation of ego-level latent classes and alter-level latent classes by introducing a nonparametric random coefficient. The main idea of this approach is to capture group-level variances by positing upper-level latent classes as well as continuous random parameters (Vermunt 2003). The benefit of this modeling strategy is not only to take into account lower-level dependency but also to generate subjectively interpretable groups of upper-level units. In this study, this non-parametric approach allows for a grouping of the respondents based on the distributional composite of subtype discussion partners. Each group of respondents can be interpreted as a group of people who have similar types of discussion partners. Thus the individual-level latent classes can be seen as distinctive types of discussion networks.

The third advantage is that the ego-level clusters derived from MLLCA can be incorporated with the regression model for estimating the effects of the latent class on an interesting outcome variable. The currently developed latent class modeling framework, the so-called bias-adjusted three-step approach, made it possible to estimate the effects of latent class on an outcome variable as regarding the latent class analysis' probabilistic nature of class membership assignment, which would otherwise lead to downward-biased estimates (Bolck et al. 2004; Vermunt 2010). As with the classic latent class analysis, MLLCA assigned upper-level units to each latent class based on the class membership's estimated probability. The bias-adjusted three-step approach uses the set of posterior classification probabilities as weight variables in the regression model for testing the effects of latent class on the outcome variables. Using this technique, I examined the interaction effect between ego-level latent class (i.e., discussion network types) and the number of negative life events on depression.

Based on the Bayesian information criterion (BIC) and group-based Bayesian information criterion (Burnham and Anderson 2004; Vermunt 2010), I selected the five-tie cluster and two-ego cluster model as the final model. The goodness-of-fit statistics are shown in Table A1 in Appendix A, with an explanation of the model selection.

### 3. Results

*3.1. Different Types of Discussion Partners*

The results from the final model suggest that there are five subtypes of discussion partners. Two types of discussion partners were well fitted with the conventional expectation that people tend to disclose their personal matters to and discuss important issues with those who are emotionally close and supportive. The other three types of discussion partners were somewhat deviant from the previous assumption in the sense that these types of discussion partners were either not intimate or less supportive. Using the conditional probabilities and posterior mean probabilities of six-tie level profiling and five-tie attribute variables shown in Table A2, I characterize the five subtypes of discussion partners. Figures 1–5 present characteristics of each type of discussion partner, respectively.

Spouse/romantic partner type: Most alters(84%) belonged to the first type of discussion partner, which I call spouse/romantic partner type, were the respondents' spouse or romantic partner with whom they lived and whom they felt very close. As shown in Figure 1, this type of alter was highly likely to be named a socializing partner and a person people would rely on when sick. It should be clear here that this type of discussion partner shows a low chance of being named as someone who helps respondents out or receives help from ego. These lower probabilities are mainly due to the restrictive survey question. UCNets asked respondents to list alters from their network members who exchange help questions but do not live with them. Thus, the lower chance of being named as exchanging help partners for this type of discussion partner does not indicate the actual absence of help exchange. Instead, the spouse/romantic partner is a primary support exchanging partner, including discussing important matters.

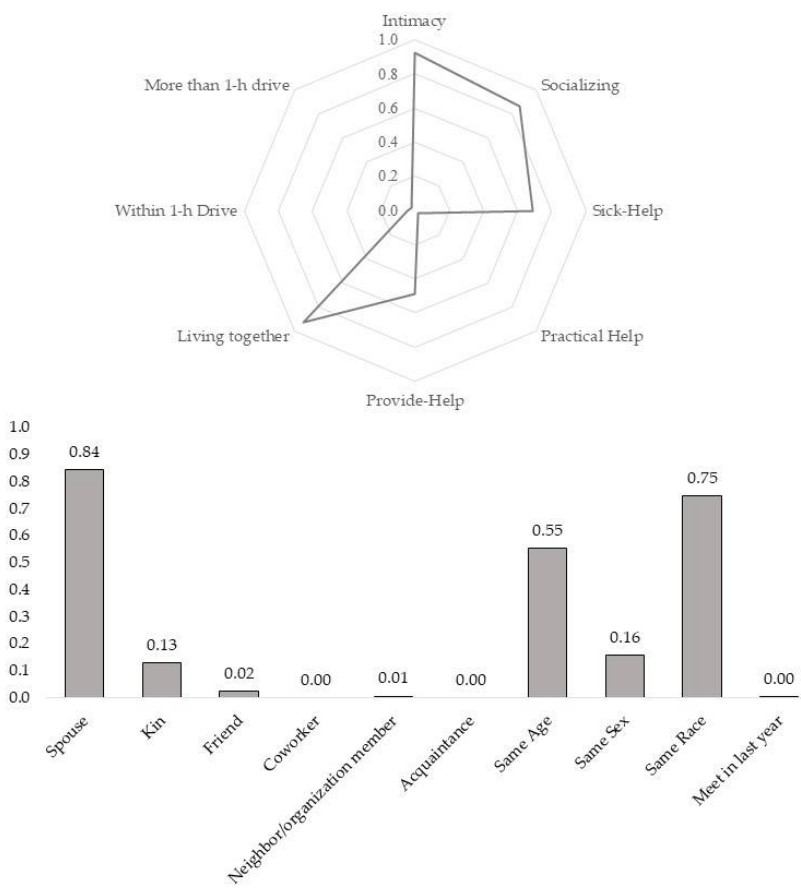

**Figure 1.** Spouse/romantic partner type.

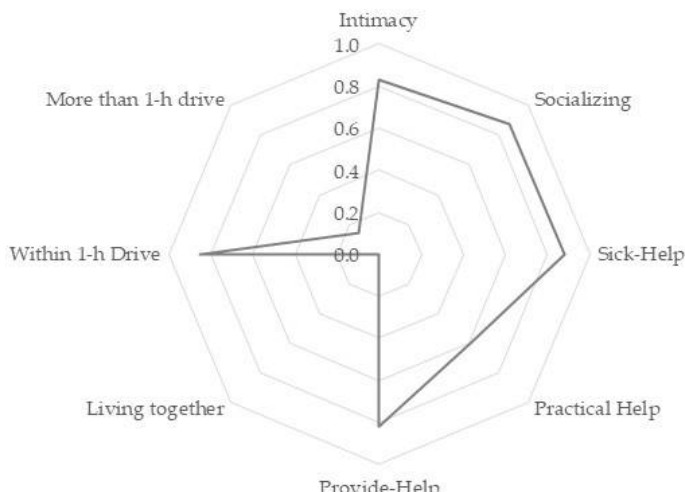

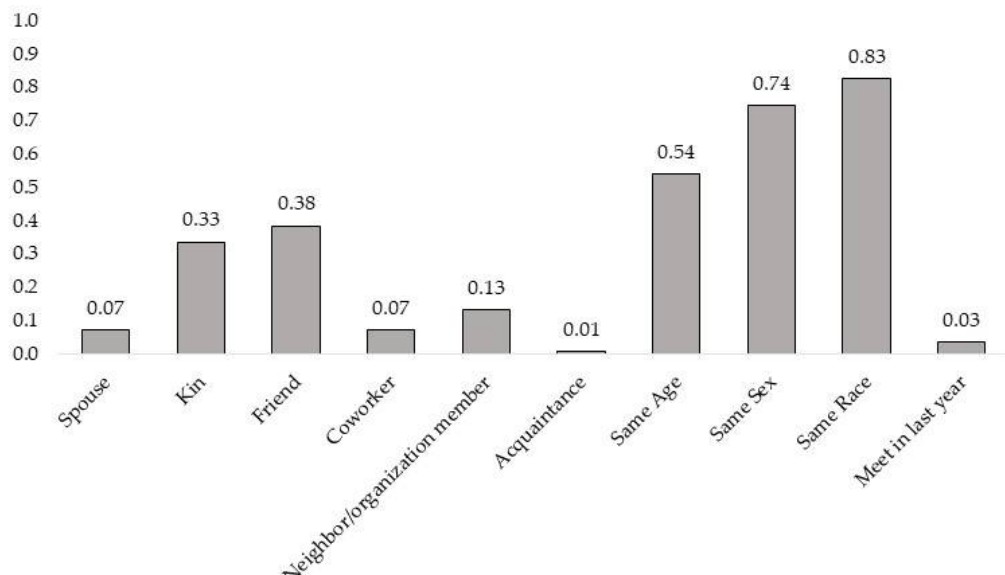

**Figure 2.** Close neighbor type.

Close neighbor type: As with the spouse/romantic type, the discussion partners assigned to this class were emotionally close, geographically reachable (living within one-hour driving distance), actively exchanged supports, and frequently socialized with the respondents. One-third of the discussion partners in this group were family members such as parents, children, siblings, or other family members, and 38% were friends. Neighbors and social organization members took up 13% of this type of discussion partner. The members of this group were likely to be homogenous in gender, race, and age. On the basis of these characteristics, I labeled this group as the close neighbor type discussion partner. This cluster made up 23.2% of all discussion partners.

Spouse/romantic partner type and close neighbor type discussion partners fit with the conventional expectation that people disclose personal matters to and discuss important issues with intimate and supportive network members. However, these two types of discussion partners only consist of 37% of all discussion partners, indicating that people select two-thirds of their discussion partners from somewhat less close or less supportive social relationships.

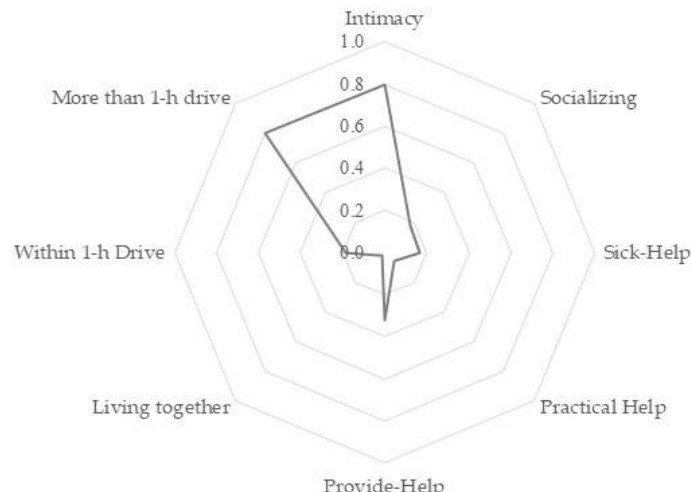

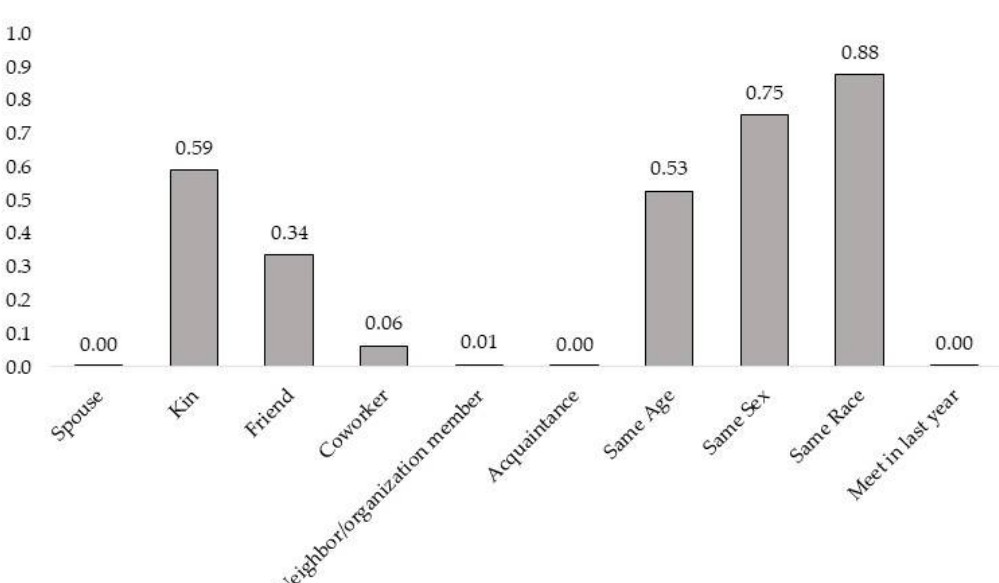

**Figure 3.** Remote type.

Remote type: Of confidant ties, 24.9% were assigned to this third class, which was named the remote type discussion partner. Most of the alters in this class lived far away from the respondent. Possibly because of this geographical separation, the respondents were less likely to mention these alters as support and socializing partners. Yet despite the geographical distance, the respondents maintained solid emotional attachment with this type of alters and sought them out for discussing important personal matters. About 60% of the alters in this class were kin members such as siblings, parents, or children (59%), and 34% were friends. As with the close neighbor type discussion partner, this group demonstrated a high level of homogeneity in race and gender.

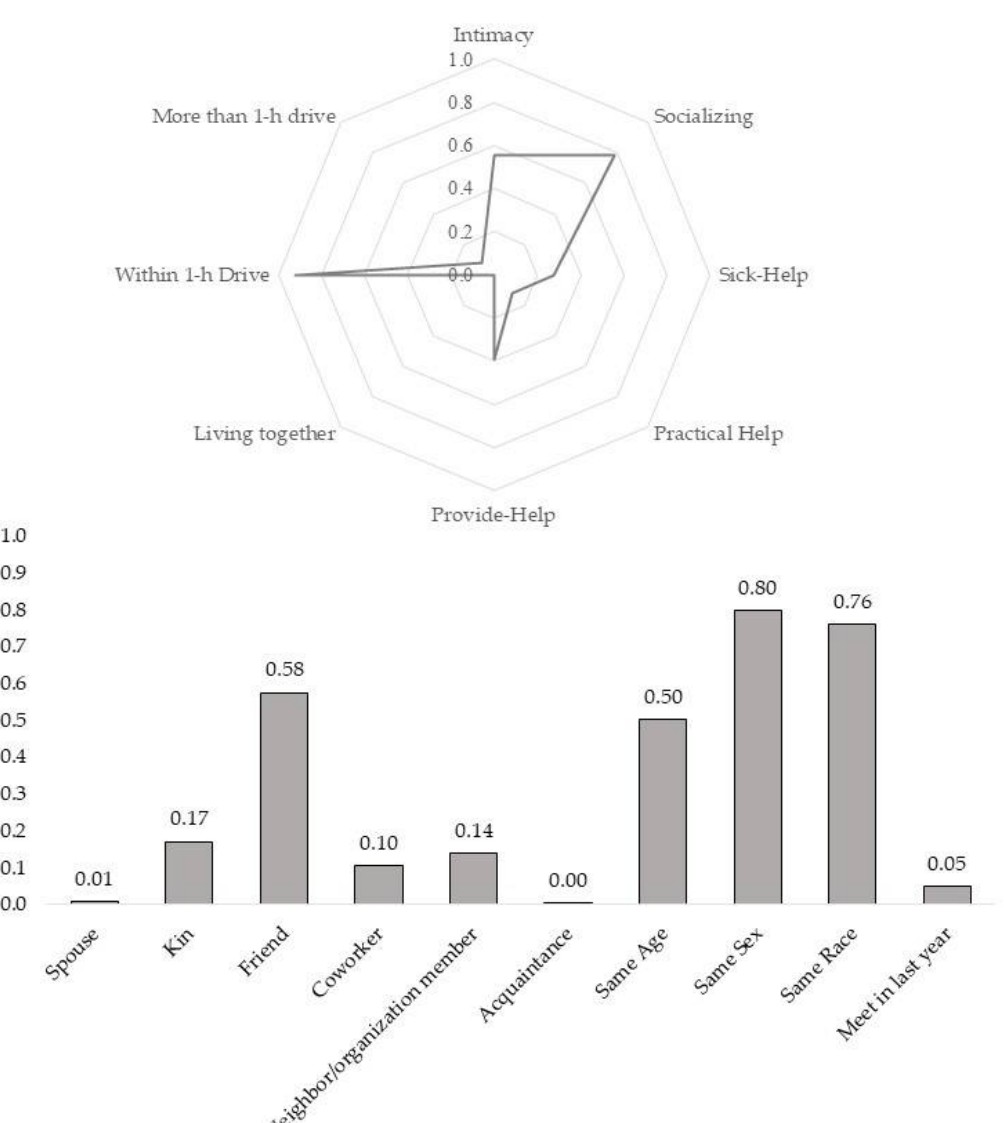

**Figure 4.** Social companion type.

Social companion type: This type of confidant made up 27.1% of overall discussion partners. Members of this class lived nearby, frequently socialized with, and were highly likely to be of the same gender and race as the respondents. However, their chance of being characterized as an especially close person was significantly lower (0.55) than the former three groups. The class-specific probabilities of the support variables—providing, receiving, and sick help—were substantially low, at 0.40, 0.12, and 0.27, respectively. This type of discussion partner can be characterized as a moderately intimate socializing partner with whom respondents are less likely to exchange help. I labeled this group of discussion partners as the social companion type. As expected, 57.5% of alters in this group were friends and coworkers. And neighbors or social organization members took up 24%. One notable characteristic of this group is that the conditional probability of newly-met alters is relatively high compared to the former three types of discussion partners. As illustrated in Table A2, 37.4% of the newly-known alters belong to this group. This type of discussion partner confirms that people sometimes disclose their personal matters to and seek advice

for important decisions from those who are not emotionally close or even someone they have known for less than one year.

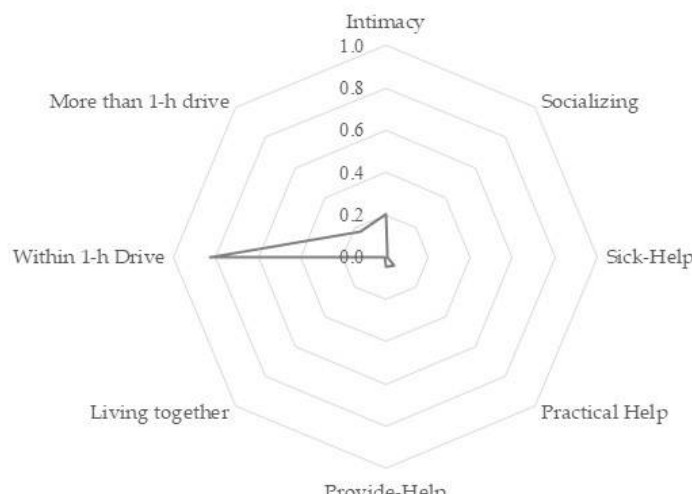

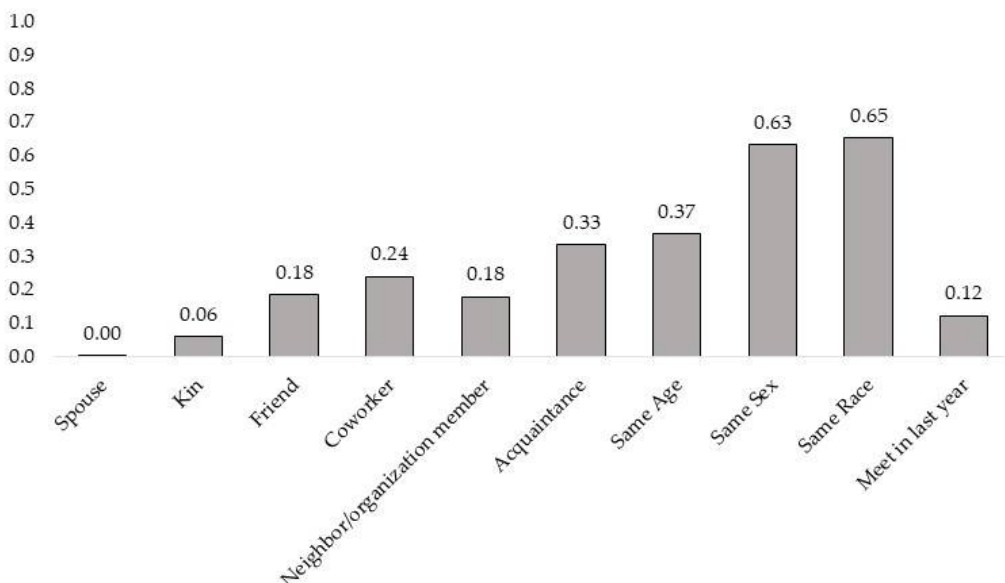

**Figure 5.** Acquaintance type.

Acquaintance type: The last type of discussion partner, which makes up 10.7% of overall discussion partners, demonstrates the most deviant characteristics from the conventional assumption about discussion partners. Alters in this class were neither emotionally close nor supportive. Although they lived within a reachable distance and played a discussion partner role, people rarely exchanged support or socialized with this type of alters. The dominant role relationships in this group were acquaintances (33.5%) and coworkers (23.9%). As illustrated in Table A2, 92.4% of alters described as acquaintances were assigned to this type. Twenty-one percent of neighbors or social organization members belong to this class. Not surprisingly, 36.3% of newly-met alters also belonged to this type of confidant. I labeled this class as the acquaintance type discussion partner.

### 3.2. Different Composition of Discussion Partners

On an individual level (level 2), the final MLLCA model indicates that the individuals were grouped into two different types of discussion networks according to the varying distribution of the five types of discussion partners. Based on each group's composition of discussion partners, I labeled the first group as Mixed Network, where 64.8% of respondents belonged, and the other one as Local Centered Network, where 36.2% were assigned. As illustrated in Table A3 and Figure 6, both discussion network types have similar percentages of spouse/romantic partner type discussion partners (17.6% and 16.4%, respectively). However, the Local Centered Network has a substantially larger number of close neighbor type discussion partners (49.3%) than the Mixed Network (6.7%). In contrast, the remote type and acquaintance type constitute a larger portion in Mixed Networks (29.5% and 17.5%, respectively) than in the Local Centered Network (9.7% and 0.0%, respectively). The Mixed Network has slightly more social companion type discussion partners than the Local Centered Networks (28.7% for Mixed Network and 24.7% for Local Centered Network).

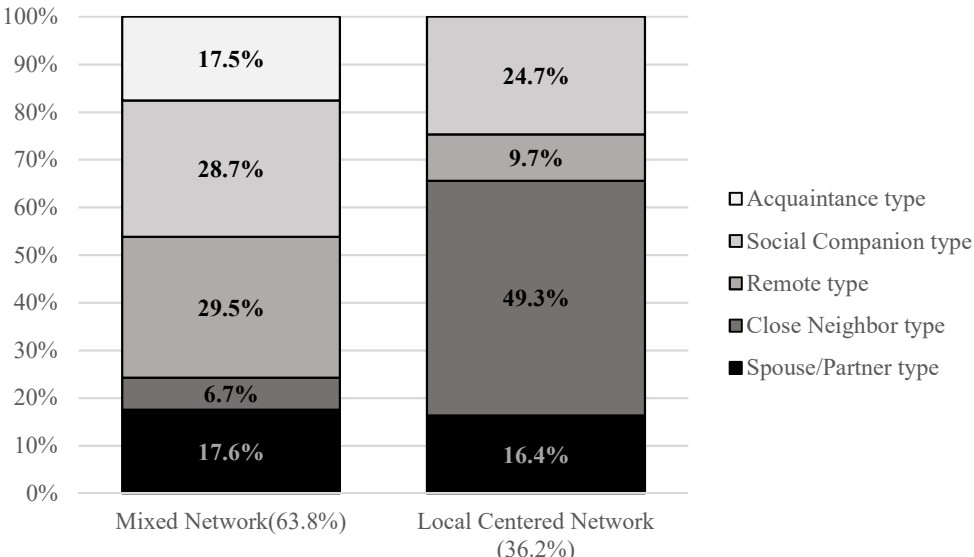

**Figure 6.** Two types of core discussion networks.

The covariate model results that tested the effects of the respondents' characteristics on the ego-level class membership showed additional differences between Mixed Network and Local Centered Network. Table 2 illustrates the estimated class proportion according to the respondent's socio-economic status and other characteristics. While health status and employment status are statistically significant, the estimated proportions of the two types of discussion networks according to these characteristics indicate that these differences are not practically substantial. Interestingly, the number of discussion partners and intimate discussion partners is not statistically and meaningfully different between the two types of discussion networks. Both types of discussion networks have about five discussion partners on average (5.13 for Mixed Network and 5.24 for Local Centered Network). The average number of close discussion partners in each discussion network was 3.3 for Mixed Network and 3.4 for Local Centered Network. Thus when there are significant variances in depression according to the type of discussion network, it can be said that the varying effect is due to the compositional differences between the two discussion network types rather than the number of discussion network members. The following section demonstrates the significant differences between the two types of discussion networks in buffering effects against the negative effects of adverse life events on depression symptoms.

**Table 2.** Proportion of respondents' characteristics by discussion network types.

| | | Mixed Network | Local Centered Network | *p*-Value |
|---|---|---|---|---|
| Class size | | 63.85% | 36.15% | |
| Mean age | | 58.965 | 59.011 | 0.91 |
| Mean number of discussion partners | | 5.128 | 5.248 | 0.82 |
| Mean number of close discussion partners | | 3.301 | 3.419 | 0.68 |
| Mean self-rated health status | | 2.489 | 2.528 | 0.0026 |
| Gender | Male | 0.630 | 0.370 | 0.083 |
| | Female | 0.627 | 0.373 | |
| Marital Status | Married | 0.675 | 0.326 | 0.91 |
| | Not married | 0.647 | 0.354 | |
| Race/Ethinicity | White | 0.647 | 0.353 | 0.22 |
| | Non-white | 0.689 | 0.311 | |
| Education level | Less than Bachelor's degree | 0.670 | 0.330 | 0.28 |
| | Bachelor's degree | 0.695 | 0.305 | |
| | More than Bachelor's degree | 0.621 | 0.379 | |
| Employment status | Not fully employed | 0.682 | 0.318 | 0.035 |
| | Fully employed | 0.647 | 0.353 | |
| Personal income | Less than $35K | 0.647 | 0.353 | 0.42 |
| | $35−$75K | 0.681 | 0.319 | |
| | More than $75K | 0.671 | 0.329 | |
| Survey method | Face-to-face | 0.690 | 0.310 | 0.022 |
| | Self-administrate | 0.590 | 0.410 | |

### 3.3. Different Effects of Discussion Partners

The above results illustrate that discussion networks are classified into two different types according to the different compositions of the five types of discussion partners. What type of discussion network would be better for mental health?

Table 3 and Figure 7 illustrate the results of the regression models with the latent variables that predict the effects of the two types of discussion networks on depression levels. Model 1 tested the direct effects of the discussion network type on depression, while Model 2 examined buffering effects of discussion networks by adding the interaction effect between the number of adverse life events and discussion network types in the model. In both models, the Mixed Network discussion network was set as a reference category for comparison.

Although the direct effect of the discussion network type is not statistically significant, as demonstrated in Model 1, the coefficient score (−0.588 in Model 1) indicates that respondents with Local Centered Network discussion networks may feel less depression than respondents with Mixed Network discussion networks. The number of total discussion partners and the size of close discussion partners are also not significantly associated with depression. Not surprisingly, the number of adverse life events is significantly associated with depression symptoms, indicating that the more people have unpleasant life events, the more depressed they feel.

**Table 3.** Effects of discussion network types on depression symptoms.

| | Model 1 | | Model 2 | |
|---|---|---|---|---|
| | coef | s.e. | coef | s.e. |
| Mixed Network discussion network | | | | |
| Local Centered Network discussion network | −0.588 | 0.414 | 0.1342 | 0.4666 |
| *N* of problems | 0.455 ** | 0.166 | 0.803 *** | 0.2306 |
| *N* of problem X Local Centered Network discussion network | | | −0.895 *** | 0.3451 |
| *N* of discussion partners | 0.000 | 0.069 | −0.004 | 0.0694 |
| *N* of close discussion partners | −0.093 | 0.084 | −0.066 | 0.0851 |

\*\* $p < 0.01$, \*\*\* $p < 0.001$. Control variables are not presented. The full model is shown in Table A4 in Appendix A.

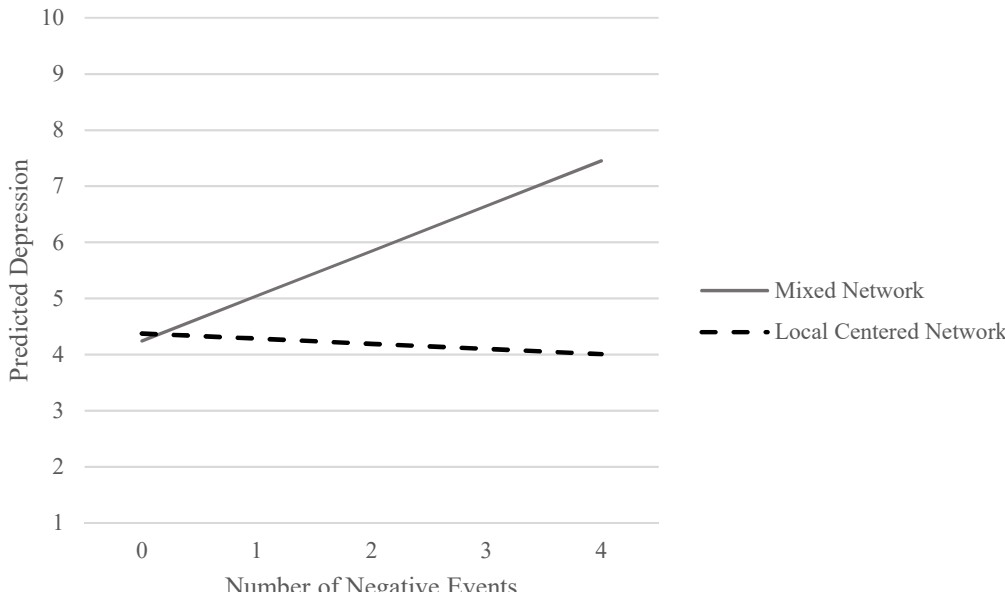

**Figure 7.** Difference between Mixed Network and Local Centered Network in buffering the adverse impact of negative life events on depression.

The interaction term between the number of adverse life events and the types of discussion network composition in Model 2 demonstrates that the Local Centered Network is better for mitigating the negative effect of undesirable life events on depression than the Mixed Network discussion network. Figure 7 plots the predicted depression along with the number of adverse events by separating the two types of discussion networks based on estimations from Model 2 (the solid line for the Local Centered Network discussion network and the broken line for the Mixed Network discussion network). Among individuals, particularly those with two or more negative events, the Local Centered Network member had substantially lower depression levels than others in the Mixed Network discussion network. For example, when everything is equal, and people experience three adverse life events, individuals in a Mixed Network type are likely to show 2.55 (=0.803*3 − (0.1342 − 0.895*3)) higher depression symptoms than others belonging to Local Centered Network types. Considering that Local Centered Network has more close neighbor type discussion partners than Mixed Network, this result suggests that the negative effect of adverse life events on depression is more effectively attenuated by discussing important personal matters with close and reachable network members than with close ties living far away(i.e., Remote type) or with weak ties in the reachable distance (i.e., social companion type and acquaintance type).

## 4. Discussion

The existing social support literature has emphasized the varying effects of social support according to the types and sources of social support. The primary goal of this study was to examine the differential effects of multiple forms of supportive social ties on depression. Instead of defining the dyadic social relationship with one specific characteristic, such as role relationship or intimacy, I proposed identifying a dyadic social tie as a multidimensional construct consisting of multiple relationship attributes. According to the configuration pattern of several attributes, dyadic social ties take different forms of relationship. The results of this study show that there are five distinct forms of discussion partner ties, from two close relationships (spouse/romantic partner types and close neighbor type) to three deviant types (remote type, social companion type, and acquaintance type). And discussion networks come together in two different ways: One group of people has more remote and acquaintance type discussion partners (i.e., Mixed Network) than others, who have more close neighbor type discussion partners (Local Centered Network). Although

the average number of discussion partners is not different between the two discussion network types, the discussion network with more close neighbor types (Local Centered Network) is better for reducing the negative influence of adverse life events on depression symptoms.

I believe that classifying multiple types of social relationships contribute to broadening our understanding of the association between social support and mental health. Social support literature has paid attention to the differential effects of social support according to the types and sources of the support (Dean et al. 1990; House 1981; Thoits 1985), but the previous empirical results are somewhat inconsistent. For example, some studies highlight the importance of a spouse relationship (Dean et al. 1990), while others report that friends play an imperative role in promoting health outcomes (Potts 1997). This inconsistency is, in part, attributable to the lack of consideration of the multifaceted aspects of dyadic social relationships. For example, the significant effect of friends on reducing depression is not just because they are friends but because they provide adequate support or entail normative pressure and regulation. In the same way, the importance of a spouse in mental well-being depends on the various characteristics of the relationship with the spouse. Depending on the several relationship aspects, the effects of friends may be similar to or different from the spouse's contribution to mental well-being. As shown in this study, some friends who belong to the close neighbor type (25% of all friends) are as close and supportive as a spouse, while other friends in the social companion type (44% of all friends) play a role as a social companion with moderate closeness. This result suggests that researchers can systemically assess the varying effects of different social relationships on depression by identifying the forms of relationships in terms of the multiple aspects of the social tie.

This study also suggests that the association between core discussion networks and health should be interpreted carefully. As studies show, core discussion partners are not necessarily strong, supportive, and stable social relationships (Bearman and Parigi 2004; Brashears 2014; Small 2013, 2017). As the results of this study demonstrate, strong and supportive confidant ties are not the dominant form of confidant relationships. The spouse/romantic partner type and close neighbor type discussion partners who provide multiple social services with strong emotional bonds and geographical proximity only constitute 37.4% of all discussion partners. More than half of discussion partners are alters who are either only specialized for social companionship (social companion type) or not physically reachable (remote type). Acquaintances who are neither close nor actively exchanging help constitute an additional 10.7% of all discussion partners. Thus, the effects of the core discussion network on health are not merely due to size. What matters is the forms of social relationships that comprise the discussion networks.

This study determined the multiple types of confidant ties and demonstrated their varying implications for mental health. There are several important issues that this study did not directly address. First, the causal links between confidant network composition and depression require further investigation. This study demonstrates that people with fewer close neighbor type confidants show higher-level depression when faced with adverse life events. This association may be because depression makes people rely more on their closest ties (e.g., spouse or romantic partner) or acquaintances such as physicians rather than their immediate network members who live nearby. Adverse life events or depression may also make the close neighbor type alters drift apart from the personal networks. The causal relationship between depression, negative life events, and selecting confidants is a promising area for future research.

Second, other relationship attributes that this study did not capture may be important elements that constitute the form of a supportive tie. As support literature suggests, appraisal support and informational support may have a different role in buffering stress (House 1981; Krause 1987). Previous studies on social support have also demonstrated that the mobilization and consequence of social support may depend on the reciprocity, obligations, and conflict aspects of social ties (Antonucci and Jackson 1990; Dean et al. 1990; Kawachi and Berkman 2001; Rook 1984). Although this study cannot account for these

characteristics due to data limitation, the provision of appraisal and informational support, reciprocity, obligation, and conflict aspects of the social relationship will contribute to a further description of certain forms of supportive ties.

Lastly, this study examined the discussion partner networks of older adults. Older adults' life circumstances and discussion topics are hardly the same as other age groups. The different life contexts are associated with varying discussion issues (Bearman and Parigi 2004) and offer different opportunities to form and develop discussion partner ties (Cornwell et al. 2008; Feld 1981; Thomas 2019). Thus, young and middle age adults' types of discussion partners would be different from older adults' discussion partners in their distribution and detail characteristics. And the buffering effects of discussion partner networks on depression would vary according to the life stages. In this sense, the generalizability of this study's findings should be carefully considered, and further assessments with different age groups made.

**Author Contributions:** As a single author, K.L. contributes all steps of the study from conceptualization, formal data analysis, writing and editing.

**Funding:** This work was supported in part by the National Institute of Aging grant R01AG041955 (principal investigator, Claude S. Fischer).

**Institutional Review Board Statement:** The study was conducted according to the guidelines of the Declaration of Helsinki, and approved by the Committee for Protection f Human Subject of the University of California Berkeley (protocol code 2012-09-4602, 10 May 2013.)

**Informed Consent Statement:** Informed consent was obtained from all subjects involved in the study.

**Data Availability Statement:** The data presented in this study are publicly available from the UCNets project website: http://ucnets.berkeley.edu/ (accessed on 12 May 2017).

**Acknowledgments:** I thank Claude Fisher, Leora Lawton, and the University of California Social Network Survey team members for their excellent comments and support. Furthermore, I appreciate Anne Pebley for her careful reading and comments. And I thank reviewers for helpful suggestions and comments.

**Conflicts of Interest:** The author declares no conflict of interest.

## Appendix A

The first five rows (M1 to M6) in Table A1 present the goodness-of-fit statistics of the latent class models that were estimated without a random coefficient. The BIC values of these models illustrate that the five-class solution (M5) is better than other models for describing the observed associations of the six relationship variables. Given this solution, the respondent-level variance was captured using parametric and nonparametric methods (Models 7 and 9). As illustrated in the BIC values of these models, both the parametric and nonparametric specifications of the respondent-level effect substantially increased the model fit compared with Model 5. Furthermore, the two-group level latent-cluster model (Model 8) illustrates the lowest BIC and Group-Based BIC. Therefore, M8 was chosen as the final solution. This model indicates that the discussion partner ties were classified into four classes, and the respondents were grouped into two clusters based on the distribution of the four confidant classes.

**Table A1.** Goodness-of-fit statistics for multilevel latent class models.

|  |  | LL | BIC | BIC-G | Parameters | Class.Err. |
|---|---|---|---|---|---|---|
| M1 | 1 Class | −10,188.17 | 20,431.07 |  | 7 | 0.00 |
| M2 | 2 Class | −9520.81 | 19,158.91 |  | 15 | 0.10 |
| M3 | 3 Class | −9359.63 | 18,899.10 |  | 23 | 0.12 |
| M4 | 4 Class | −9263.64 | 18,769.66 |  | 31 | 0.14 |
| M5 | 5 Class | −9226.75 | 18,758.43 |  | 39 | 0.22 |
| M6 | 6 Class | −9213.63 | 18,794.75 |  | 47 | 0.25 |
| M7 | 5 Class + 1 Gclass + r | −8247.22 | 17,112.10 | 16,995.61 | 79 | 0.14 |
| M8 | 5 Class + 2 Gclass + r | −8221.21 | 17,099.16 | 16,975.30 | 84 | 0.14 |
| M9 | 5 Class + 3 Gclass + r | −8215.97 | 17,127.79 | 16,996.55 | 89 | 0.14 |

Note: BIC = Bayesian information criterion; BIC-G = group-based Bayesian information criterion; Class.Err. = mean of the proportion of classification error for the latent class; Gclass = respondent level cluster; r = random parameter.

**Table A2.** Conditional probabilities and profiles for five types of discussion partner ties.

| Conditional Probabilities | Spouse/Romantic Partner Type (13.9%) | Close Neighbor Type (23.5%) | Remote Type (24.9%) | Social Companion Type (27.1%) | Acquaintance Type (10.7%) |
|---|---|---|---|---|---|
| Intimacy | 0.923 | 0.830 | 0.798 | 0.554 | 0.206 |
| Socializing | 0.867 | 0.878 | 0.174 | 0.787 | 0.017 |
| Sick help | 0.691 | 0.880 | 0.165 | 0.277 | 0.010 |
| Practical help | 0.026 | 0.603 | 0.060 | 0.117 | 0.056 |
| Provide help | 0.487 | 0.819 | 0.322 | 0.389 | 0.048 |
| Living together | 0.928 | 0.004 | 0.017 | 0.000 | 0.003 |
| Within 1-h drive | 0.045 | 0.855 | 0.183 | 0.922 | 0.828 |
| More than 1-h drive | 0.027 | 0.141 | 0.800 | 0.078 | 0.169 |
| Profiles: Role relationships, similarities, and newly met relationship | | | | | |
| Spouse/romantic partner | 0.8596 | 0.1241 | 0.0001 | 0.0132 | 0.003 |
| Kin | 0.0597 | 0.2654 | 0.4964 | 0.1567 | 0.0218 |
| Friend | 0.009 | 0.2554 | 0.2374 | 0.4426 | 0.0555 |
| Coworker | 0.000 | 0.194 | 0.1838 | 0.3277 | 0.2944 |
| Neighbor/organization member | 0.0078 | 0.3426 | 0.0213 | 0.4169 | 0.2115 |
| Acquaintance | 0.0000 | 0.0394 | 0.0104 | 0.0264 | 0.9237 |
| Same age | 0.1507 | 0.2485 | 0.2576 | 0.2664 | 0.0768 |
| Same sex | 0.0323 | 0.2615 | 0.281 | 0.3243 | 0.101 |
| Same race | 0.131 | 0.2448 | 0.2756 | 0.2607 | 0.0878 |
| Met in last year | 0.0136 | 0.2179 | 0.0315 | 0.3744 | 0.3625 |

**Table A3.** Distribution of five types of discussion ties.

|  | Mixed Network (63.8%) | Local Centered Network (36.2%) | *p*-Value |
|---|---|---|---|
| Distribution of five types of discussion partners | | | |
| Spouse/partner type | 17.6% | 16.4% |  |
| Close neighbor type | 6.7% | 49.3% |  |
| Remote type | 29.5% | 9.7% |  |
| Social companion type | 28.7% | 24.7% |  |
| Acquaintance type | 17.5% | 0.0% |  |

**Table A4.** Effects of discussion network types on depression symptoms: full model.

| | Model 1 | | Model 2 | |
|---|---|---|---|---|
| | **Coef** | **s.e.** | **Coef** | **s.e.** |
| Mixed Network discussion network | | | | |
| Local Centered Network discussion network | −0.588 | 0.414 | 0.1342 | 0.4666 |
| *N* of problems | 0.455 ** | 0.166 | 0.803 *** | 0.2306 |
| *N* of problem X Local Centered Network discussion network | | | −0.895 *** | 0.3451 |
| *N* of discussion partners | 0.000 | 0.069 | −0.004 | 0.0694 |
| *N* of close discussion partners | −0.093 | 0.084 | −0.066 | 0.0851 |
| Age | −0.147 *** | 0.024 | −0.156 *** | 0.0228 |
| Male | −0.155 *** | 0.023 | | |
| Female | −0.156 | 0.275 | −0.197 | 0.2744 |
| Married | | | | |
| Not married | −0.225 | 0.371 | −0.248 | 0.3725 |
| White | | | | |
| Non-white | −0.118 | 0.279 | −0.102 | 0.2746 |
| Less than Bachelor's degree | | | | |
| Bachelor's degree | −0.482 | 0.300 | −0.558 | 0.3004 |
| More than Bachelor's degree | 0.015 | 0.328 | 0.006 | 0.3258 |
| Not fully employed | | | | |
| Fully employed | −0.146 | 0.301 | −0.12 | 0.297 |
| Less than 35K | | | | |
| 35−75K | −0.868 | 0.509 | −0.891 | 0.5061 |
| More than 75K | −1.943 *** | 0.512 | −1.950 *** | 0.5146 |
| Self-rated health status | 1.178 *** | 0.147 | 1.161 *** | 0.1468 |
| Self-administered survey | 0.457 | 0.317 | 0.4387 | 0.3121 |
| Constant | 12.930 *** | 1.631 | 12.696 *** | 1.5998 |

** $p < 0.01$ *** $p < 0.001$.

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
