# Peer review of "Different Discussion Partners and Their Effect on Depression among Older Adults"

_socsci, doi:10.3390/socsci10060215_

Round 1

Reviewer 1 Report

See attached document with recommended edits and questions. Overall, excellent study. One criticism is that your title and abstract are misleading as your findings are not quite as generalizable as you suggested. Your participant pool was older adults and this should be indicated. such a finite sample requires limited interpretation of the results. You need to add to your discussion section comments on the limitations of your study due to a limited sample.

Reviewer 2 Report

This paper uses UCNets data to (a) conceptualize and measure the multidimensionality of social network relationships, (b) identify network typology, and (c) assess the moderating effects of networks on depression. Unlike previous studies, the paper goes beyond unidimensional measures of relationships to consider multidimensional aspects. Overall, the paper is well written and makes a clear contribution to the social network literature. Please see below for my comments.  

  1. The author calls first mention a social support network (p.1, line 3) but latter calls it a core discussion network, the latter of which is more accurate given the name generators they choose to analyze.

INTRODUCTION

  1. Within the first few pages it was unclear to me whether the author was proposing a tie-level analysis or ego-level analysis. I assumed it was the latter since the outcome is depression but it was not until quite late in the manuscript (roughly p.4) that it became clear to me that they would be taking a multi-step approach in which they used a typology approach at the tie-level and then moved to the ego-level analysis. Might want to make this clear earlier.

  1. To my previous point, the first hypothesis (though not labeled as H1) on p.4, lines 200-202 appears to combine elements of tie-level predictions (e.g., “there will be diverse discussants who offer distinctive forms of relationship in terms of intimacy, accessibility, and support provision…”) and network-level predictions (e.g., “…and that the core discussion networks will be differently configured according to the distribution of multiple forms of discussants”). The author should consider separating this statement into two clearly marked different hypotheses, along with the final hypothesis relating to depression (p.4, lines 203-205).

DATA AND METHODS

  1. Why did the author choose to only focus on older adults if UCNets sampled a younger cohort as well? Some theoretical explanation earlier in the manuscript would be helpful in explaining why this paper focuses on older adults. To this end, I also suggest that this is mentioned in the Discussion of how the highlighted relationship types in this manuscript might (or might not) be unique to later life.

  1. It would be helpful to the reader to list the six profiling variables (p.5, line 241) in the text when first discussing them. I was confused as to which six variables these referred to, nor did I initially see them in Table 1. The author says that these six variables (along with the 5 alter attributes) are listed in the left and right panels of Table 1 but from what I can tell, they all appear in the right panel of Table 1. In the next subsection (‘Alter-level variable’) the author mentions three main relationship components (intimacy, geographic reachability, and social exchanging), not six. It looks like social exchanging gets broken down into three variables. Are these the six tie-level variables the author was previous referring to on p.5, line 241? It is not clear, especially since the author previously called them tie-level variables (p.5, line 241), but now calls them alter-level variables (p.5, line 246). Please clarify.

  1. Please reorder the first mention of the list of relationship components (p.5, line 248) to match the subsequent description of these components (p.5, lines 248-255). 

  1. More to the point of the inclusion of these alter-level variables, I don’t see any justification for including them in the analysis. Some seem relatively random for your purposes (e.g., first met in last year). I would suggest including either theoretical justification or, at minimum, citing some studies that have considered these variables when assessing relationship/network type.

REUSLTS

  1. It would be helpful to the reader if you assign meaningful names to the network types rather than just calling them Type I and Type II. I kept forgetting which type was which.

  1. On p.6, line 282, the author says that they will be “testing the mediate effects of discussion partner networks in the association between the number of negative events and depression symptoms.” But this is not what they actually test. The interaction models (along with talk of buffering) suggest that they are actually testing moderation. This is fine but they need to clarify the language throughout the manuscript so as not to misdirect the reader.

  1. Regarding the discussion of the findings from the regression models in Table 3, I would appreciate mention of numeric values to help give the reader a sense of the magnitude of the associations. Simply using terms like “higher” and “lower” do not get at whether this is a relationship of substance. This could be done by verbalizing the marginal effects that were shown in Figure 7.

MINOR NOTES

  1. Average network size listed on p.5, line 239 (5.25) does not match the value provided in Table 1 (5.26). I imagine this was a rounding error.

Round 2

Reviewer 2 Report

After reviewing the revisions to manuscript socsci-1191253, I recommend that this manuscript should be accepted for publication by Social Sciences. The author has done well to address my comments and has produced a quality manuscript as a result.

Sincerely,